# *DhDIT2* Encodes a *Debaryomyces hansenii* Cytochrome P450 Involved in Benzo(a)pyrene Degradation—A Proposal for Mycoremediation

**DOI:** 10.3390/jof8111150

**Published:** 2022-10-30

**Authors:** Francisco Padilla-Garfias, Norma Silvia Sánchez, Martha Calahorra, Antonio Peña

**Affiliations:** Departamento de Genética Molecular, Instituto de Fisiología Celular, Universidad Nacional Autónoma de México, Circuito Exterior s/n, Ciudad Universitaria, Mexico City 04510, Mexico

**Keywords:** *Debaryomyces hansenii*, yeast, benzo(a)pyrene, mycoremediation

## Abstract

Pollutants, such as polycyclic aromatic hydrocarbons (PAHs), e.g., benzo(a)pyrene (BaP), are common components of contaminating mixtures. Such compounds are ubiquitous, extremely toxic, and they pollute soils and aquatic niches. The need for new microorganism-based remediation strategies prompted researchers to identify the most suitable organisms to eliminate pollutants without interfering with the ecosystem. We analyzed the effect caused by BaP on the growth properties of *Candida albicans, Debaryomyces hansenii, Rhodotorula mucilaginosa*, and *Saccharomyces cerevisiae*. Their ability to metabolize BaP was also evaluated. The aim was to identify an optimal candidate to be used as the central component of a mycoremediation strategy. The results show that all four yeast species metabolized BaP by more than 70%, whereas their viability was not affected. The best results were observed for *D. hansenii*. When an incubation was performed in the presence of a cytochrome P450 (CYP) inhibitor, no BaP degradation was observed. Thus, the initial oxidation step is mediated by a CYP enzyme. Additionally, this study identified the *D. hansenii DhDIT2* gene as essential to perform the initial degradation of BaP. Hence, we propose that *D. hansenii* and a *S. cerevisiae* expressing the *DhDIT2* gene are suitable candidates to degrade BaP in contaminated environments.

## 1. Introduction

The environmental pollution caused by hazardous materials increased in recent years. The polycyclic aromatic hydrocarbons (PAHs) are among the pollutants increasingly found in nature. They are derived from the combustion of more than 100 different components of fossil fuels, but they are also produced by natural processes and anthropogenic activities [1,2]. The effects caused by these compounds on human health are long known. In 1761, the physician John Hill detected an increased incidence of nose cancer among workers involved in soot removal, and subsequently in tobacco smokers [3].

Benzo(a)pyrene (BaP) is the main representative of the PAHs. It has a five-ring structure, it occurs in solid form, and it is a highly hydrophobic compound. It is frequently found in the air, on the surface of contaminated water, as well as in sediments and soils [4,5,6]. Its levels are high in waste products derived from coal processing, oil sludge, asphalt, food products, and tobacco smoke [5,7]. It was shown that BaP and other PAHs result from the combustion of fossil fuels, wood, and other organic materials [5]. Thus, human exposure to BaP is very common and it is mainly generated by anthropogenic sources rather than natural emissions [5,7]. The highly hydrophobic nature of BaP and other PAHs cause their accumulation [5]. The United States Environmental Protection Agency (US EPA) reported the BaP half-life values for several environments: 1–6 days in the atmosphere, 1–8 h in water, 5–10 years in sediments, and 14–16 months in soils [8]. The European Union (EU) established a BaP limit value of 1 ppb in the air [9]. In this regard, a 14 ppb level was detected in Genoa, Italy [5,10], and the soils in Cleveland, USA, contain up to 5.5 ppm BaP [5], whereas the content of BaP and overall PAHs within the rivers of India are 8.6 ppb and 158 ppb, respectively [5,11]. The maximum BaP level permitted by the US EPA in drinking water is 0.2 ppm [12]. Exposure to BaP peaked in recent years because an increased contact with several emission sources and through food [5]. In mammals, BaP is readily absorbed through nasal, oral, and cutaneous routes [5,7]. After its activation by CYP-mediated metabolism it becomes a highly toxic and carcinogenic compound [4,5,7,12]. It was reported that BaP is epigenotoxic, neurotoxic, teratogenic, potentially prooxidative, and it impairs fertility in animals [2,4,5,7,12].

The elimination of these toxic compounds represents a great challenge and bioremediation may be a strategy to achieve it. Bioremediation harnesses microbial properties to counteract the damage caused to ecosystems by the presence of pollutants [2,4,13]. Most studies on PAH degradation focused on bacteria and algae, but t overlooked fungi, particularly yeasts; a group of organisms that may provide mycoremediation [3,13]. The yeasts studied so far for mycoremediation purposes belong to the Ascomycota and Basidiomycota phyla, and they were isolated from hydrocarbon-contaminated sites [13,14,15,16]. The role of the microbial populations in such sites needs to be elucidated in order to optimize the removal of PAHs in other environments, [17].

It is known that PAH metabolism depends on the ecological niche and the nutritional conditions for the organisms [4,13]. Fungi possess diverse intracellular and extracellular enzyme systems that metabolize multiple xenobiotics, and they overall comprise the xenoma [3,13]. According to some reports, fungi from the Basidiomycota phylum, such as *Phanerochaete chrysosporium*, produce laccase, lignin peroxidase, and manganese peroxidase enzymes under ligninolytic conditions or in the presence of lignocellulosic substrates, such as wood (as it contains a lignin, phenolic polymer). These enzymes oxidize lignin, and they may also oxidize PAHs [2,3,13,15]. Conversely, the fungi belonging to the Ascomycota phylum, such as yeasts, do not secrete ligninolytic enzymes and they only oxidize PAHs intracellularly [13,17,18].

PAH metabolism is divided into three steps (Figure 1) [13]. The metabolic pathway proposed by several authors establishes that PAHs enter the cell through a mechanism that remains to be elucidated, then the cytochrome P450 (CYP) monooxygenases catalyze a one-ring epoxidation to produce an unstable arene oxide that is subsequently transformed to a trans-dihydrodiol through a reaction catalyzed by an epoxide hydrolase (EH). The arene oxide produced by CYP may also be rearranged by non-enzymatic reactions to produce phenol derivatives that are subsequently conjugated with glutathione, sulphate, xylose, glucuronic acid, or glucose moieties by transferase enzymes. These conjugates may be either stored in vacuoles, mineralized, or they are secreted to be mineralized by other organisms [2,3,13,15,17,18,19,20,21]. One of the most important enzymes that contribute to the degradation process is CYP, as it catalyzes the initial oxidation of PAHs [18,19,20,21].

In 1973, Dehnen et al. characterized a benzo(a)pyrene hydroxylase in the *Saccharomyces cerevisiae* yeast. Such an enzyme was later identified as a member of the CYP superfamily [22]. Recently, Ostrem-Loss et al. confirmed that a CYP carry out the initial PAHs metabolism. They identified a CYP linked to BaP metabolism in *Aspergillus nidulans.* Such an enzyme is encoded by the *AN1884* gene (also termed *bapA*), and it is translated into the CYP617D1 protein [18,23]. Research on fungal genomes identified more than 600 genes encoding putative CYPs. The diversity of the CYP family as identified in ascomycete fungi appears to be much higher when compared to that of basidiomycetes [13,24,25]. Because of their high ability to hydroxylate complex hydrocarbons, such as PAHs, several efforts are focused on using CYPs as biocatalysts. A particular strategy involves their overexpression in selected organisms to optimize biodegradation processes [25].

In this work, the growth of *Debaryomyces hansenii* was evaluated in the presence of BaP. This yeast is osmotolerant and oleaginous, and it is commonly found in habitats distinguished by low water activities [26,27,28]. Due to its genetic and biochemical diversity, the extremophile *D. hansenii* is considered a novel reservoir of enzymes with potential technological applications, including the degradation of PAHs and other pollutants for environmental preservation purposes [27]. Our results demonstrate that BaP degradation depends on its concentration and the amount of yeast in a detoxification process probably mediated by CYP. Furthermore, we observed that the *D. hansenii DhDIT2* gene is vital for BaP metabolism and possibly for other PAHs as well. The results observed for the heterologous expression of the *DhDIT2* gene in a *S. cerevisiae dit2Δ* background confirmed that this gene participates in the degradation of BaP. Finally, the yeast *D. hansenii* is proposed as an organism suitable for mycoremediation purposes that may be used in polluted oceans by virtue of its marine origin. 

## 2. Materials and Methods

### 2.1. Yeast Strains and Growth Conditions

We used the following strains: *Debaryomyces hansenii* Y7426 (from the US Department of Agriculture, Peoria, Illinois, USA), *Candida albicans* ATCC 10231, *Saccharomyces cerevisiae* BY4742 (*MATα his3Δ1 leu2Δ0 lys2Δ0 ura3Δ0*) and its isogenic *dit2Δ* mutant (*MATα his3Δ1 leu2Δ0 lys2Δ0 ura3Δ0 dit2Δ::kanMX*) (from the yeast knockout collection, kindly donated by Dr. Gabriel del Río, IFC, UNAM), *Saccharomyces cerevisiae* W303 (*MATa/MATα* {*leu2-3,112 trp1-1 can1-100 ura3-1 ade2-1 his3-11,15*} [*phi^+^*]), and *Rhodotorula mucilaginosa* ATCC 66034 (donated by Dr. Salvador Uribe from IFC, UNAM). All strains were preserved in solid 0.67% YNBG medium (yeast nitrogen base containing amino acids and ammonium sulphate), supplemented with 2% glucose and 2% agar. All cultures were renewed once every month. The cells from the YNBG plates were routinely placed in 500 mL of YNBG medium and incubated for 24 h in 1 L-Erlenmeyer flasks placed on orbital shakers at 250 rpm and 28–30 °C. The cells were collected and washed twice with water by centrifugation. The final pellet was weighed and suspended in water at a 1:1 ratio. Cells were kept on ice until their use on the same day. 

### 2.2. Growth Assays

All steps of the growth assays were performed under aseptic conditions. 

Growth curves were obtained in YNBG medium using a 24 h yeast pre-culture suspension at 0.03 OD_600_ (optical density at 600 nm) as measured in a Beckman DU 650 spectrophotometer. The effect of several BaP concentrations (10 to 500 ppm) on growth was evaluated in a YNB medium for 48 h at 1 h intervals at 600 nm and 28 °C using a Bioscreen C automated plate reader (in this equipment the initial reading of OD_600_ is around 0.2).

Growth was studied by spotting 10-fold serial dilutions of the respective yeast suspensions on plates. Cells were previously grown for 24 h in YNBG medium, they were washed twice with sterile water and a working suspension was prepared at OD_600_ = 1.0. Dilutions were made with sterile water in a 96-well plate. Approximately 5 µL of each dilution was spotted on the surface of YNB plates supplemented with several BaP concentrations by using an aluminum multi-pin device. Controls were monitored in YNB, YNBG, and YNB media containing acetone (a solvent used to solubilize BaP in a volume equivalent to the highest BaP concentration). Incubation temperatures were 4, 15, 28, and 37 °C. The plates were incubated at 15, 28, and 37 °C and they were photographed after 72 h, whereas that incubated at 4 °C was photographed after one month because of its different growth rate.

### 2.3. Cell Viability

To quantify whether BaP affected cells long term, their viability was measured by using an alamarBlue^TM^ commercial kit (Invitrogen REF A50100, Carlsbad, CA, USA). Briefly, 24 h cell cultures grown in YNBG medium were harvested and resuspended at OD_600_ = 0.2 in either YNB or YNB media supplemented with 2% glucose or with 100 ppm benzo(a)pyrene. Then, 90 µL samples were collected at days 1, 2, 3, 5, 7, and 10 (according to the BaP degradation curve), and they were diluted at OD_600_ = 0.1 using their respective medium. Subsequently, 10 µL of the alamarBlue^TM^ reagent were added, and all samples were incubated for 4 h at 28 °C. Fluorescence (at 530 and 590 nm excitation and emission wavelengths, respectively) was measured in 96-well plates in a FLUOstar Omega plate reader. Controls containing no cells were simultaneously run.

### 2.4. Benzo(a)pyrene Degradation 

A 50 mL volume of 24 h cell pre-cultures grown in YNBG medium was used to inoculate a new flask containing 250 mL of YNB supplemented with 100 ppm BaP at OD_600_ = 0.1. A 3 mL aliquot was collected at 0, 1, 2, 3, 5, 7, and 10 days, and it was kept frozen at −20 °C to subsequently extract and quantify BaP. To extract the non-degraded BaP, the aliquots were thawed, 3 mL of chloroform were added, and they were vigorously shaken for 5 min in 15 mL conical polypropylene tubes. These were then centrifuged at 3100 rpm (1625× *g*) in a clinical centrifuge to promote a phase separation. The organic layer was recovered using a syringe and this step was performed twice to have a 9 mL final volume. Chloroform was evaporated at 62 °C inside a fume hood, the remaining BaP contained in the tube was solubilized with 3 mL of acetone, and it was finally kept at 4 °C. BaP was quantified in water by fluorescence at 356–405 nm (excitation and emission wavelengths, respectively) using an Olis-converted SLM AMINCO spectrofluorometer at room temperature using based on a standard curve. The degradation percentage was calculated from the remaining BaP contained in each tube. A control with no cells was considered as the 100% BaP level, i.e., 0% degradation. In order to confirm that BaP degradation was mediated by the cytochromes P450 (CYP), we included 10 mM piperonyl butoxide (PBO) and a specific inhibitor of CYPs in the culture medium, and the same quantification procedure was performed. The remaining BaP was extracted after a 10-day incubation and the percentage of degradation was reported. An additional control was included: a 50% (*w*/*v*) yeast suspension submitted to boiling for 20 min (dead cells) that was subsequently treated as all the other samples. 

To confirm BaP degradation by *D. hansenii* and the other three yeasts, we analyzed pooled extracts from the 10th day in the presence or absence of PBO, as well as the short-term degradation samples (see below, Section 2.5) by gas chromatography coupled to mass spectrometry (GC-MS). Additionally, a 5-point BaP standard curve (20, 100, 200, 400, 500 μM or 5, 25, 50, 100, 130 ppm) was used to quantify the remaining BaP. The data were then fitted to an exponential decay model: Q_t_ = Q_o_e^-kt^. A GC-MS was performed in a Hewlett-Packard 5890-II instrument coupled to a JEOL SX-102 A spectrometer. The GC conditions were the following: 20 HP-5MS (5% phenyl) methylpolysiloxane column (30 × 0.25 mm, Agilent Technologies, Santa Clara, CA, USA), 0.25 μm film thickness, a 30 cm/s He linear velocity, 50 °C isothermal for 3 min, linear gradient up to 300 °C at a 20 °C/min rate, 10 min hold at the final temperature. MS conditions were the following: 70 eV ionization energy, ion source temperature 280 °C, interface temperature 300 °C, scan speed at 2 scans/s, and a 33–880 amu mass range.

### 2.5. Genetic Analysis and Heterologous Expression of a D. hansenii CYP Linked to benzo(a)pyrene Degradation 

As the degradation of PAHs by organisms was primarily reported for lignocellulosic and filamentous fungi, such as *A. nidulans*, we performed a query in the AspGD database (http://www.aspgd.org) to identify the *AN1884* gene (*bapA*) that codes for CYP617D1 [18]. Two databases were queried to identify CYP homologues in *C. albicans, D. hansenii, R. mucilaginosa,* and *S. cerevisiae:* Cytochrome P450 (https://drnelson.uthsc.edu/Cytochrome450.html, accessed on 8 June 2020) [29] and Fungal cytochrome P450 (http://p450.riceblast.snu.ac.kr/index.php?a=view, accessed on 8 June 2020) [30]. Specifically, to identify the *AN1884* homologues in *D. hansenii* and in *S. cerevisiae*, other databases (https://fungi.ensembl.org/index.html, accessed on 9 June 2020 [31] and https://www.yeastgenome.org, accessed on 10 June 2020 [32]) were used.

The *DEHA2C02596g* gene (1575 bp) was identified in *D. hansenii*. It encodes the CYP52A44 protein (525 amino acid residues), a homologue of the *S. cerevisiae DIT2* gene. The amino acid sequences of the proteins involved in PAH metabolism in different yeasts [33] were aligned using the Clustal Omega platform (https://www.ebi.ac.uk/Tools/msa/clustalo/ accessed on 22 June 2020).

The *S. cerevisiae dit2Δ* mutant and its BY4742 parent strain were used for heterologous expression of the *DEHA2C02596g* gene (now termed *DhDIT2).*
Figure 2 depicts the plasmid construction based on pYES2, whereas the used primers are shown in Table 1. 

Briefly, based on the *DhDIT2* (ORF *DEHA2C02596g*) cDNA sequence, no CTG codons were identified (*D. hansenii* belongs to the CTG clade [34]), thus a regular DNA manipulation was performed. The full-length coding region was amplified by a polymerase chain reaction (PCR) using the dhdita and dhditb primers. Then, BamH1 and Not1 restriction sites were introduced by PCR using the dh450a and dh450c primers. After the amplification, the PCR products of the expected size (∼1.6 kb) were purified (QIAquick PCR and Gel Purification Kit, Qiagen, Hilden, Germany). Both, the PCR products and the pYES2 vector (Invitrogen, Carlsbad, CA, USA) were digested with BamHI-HF and Not1-HF (NEB, Ipswich, MA, US) at 37 °C for 30 min, and the digestion reactions were purified using a Monarch^®^ PCR DNA Cleanup Kit (NEB, Ipswich, MA, USA) to be subsequently ligated using a Rapid DNA Ligation Kit (Roche, Mannheim, Germany). A competent *Escherichia coli* DH5α strain was used to propagate plasmids using the thermal shock technique. Bacteria were grown at 37 °C in LB (1% tryptone, 0.5% yeast extract, 0.5% NaCl, pH 7.0) containing 100 μg/mL ampicillin for plasmid selection. Ampicillin-resistant *E. coli* colonies were grown overnight in liquid LB medium at 37 °C and their plasmids were extracted using the QIAprep Spin Miniprep Kit (Qiagen, Hilden, Germany). All constructs were validated by double digestion as mentioned above.

The double digestion positive plasmids were used to transform the *S. cerevisiae dit2Δ* mutant and its wild-type strain BY4742. Shortly, an overnight 100 mL culture in YPD medium (10% yeast extract, 20% peptone, and 20% glucose) was inoculated with the *dit2Δ* mutant and the BY4742 strain at OD_600_ = 0.3–0.4. The culture medium was then centrifuged (1700× *g* for 5 min) and washed twice with water. Both strains were transformed using the lithium acetate method [35,36] and positive clones were selected in a synthetic medium without uracil containing 2% galactose. The correct insertion of *DhDIT2* was verified by PCR (using the dhdita and dhditb primers, Table 1). The wild-type strain was also transformed with the plasmid harboring the *DhDIT2* gene to verify whether a double gene loading of *DIT2* may exhibit a significant phenotype. Moreover, the wild-type *S. cerevisiae* and the mutant lacking the *dit2* gene were transformed as previously described using the empty plasmid and they were used as controls.

To check the correct transformation of the strains and the resulting phenotype, the wild-type and transformed strains were serially diluted (as previously described) in a YNB selection medium containing 2% galactose without uracil, and in YNB medium containing uracil, 0.2% galactose, and 100 ppm BaP. 

To evaluate the short-term degradation (24 h) of BaP, 500 mL pre-cultures were inoculated with the wild-type and transformed strains grown in YNB medium containing either 2% glucose or 2% galactose. After 24 h, the cells were harvested, washed, and 1 g of biomass (wet weight) was incubated in a 20 mL volume of YNB medium supplemented with 100 ppm BaP (containing either 0.2% glucose or 0.2% galactose) at OD_600_ = 5.5. After 24 h, the remaining non-degraded BaP was extracted and quantified as previously mentioned.

### 2.6. Cytochrome P450 Reductase (CPR) Activity

CYP activity was indirectly quantified by measuring the CPR enzyme (NADPH-cytochrome *c* reductase) in both control and transformed yeast strains, either in the absence or presence of BaP. The activity was measured using a commercial kit (Cytochrome *c* Reductase (NADPH) Assay Kit CY0100, Sigma, Saint Louis, MO, USA). Briefly, cells were cultured in YNBG (control) and YNB media supplemented with 100 ppm BaP. Samples were collected at days 0, 1, 2, 3, and 6 (according to the BaP degradation curve) and then they were submitted to centrifugation. The pellets were resuspended at 1 g/mL (wet weight/volume) in 0.6 M D-mannitol, 5 mM MES (2-(N-morpholino) ethanesulfonic acid), pH 6.8 (adjusted with triethanolamine), and finally lyticase was added (150 units/g, wet weight). Spheroplasts were obtained after a 45 min incubation at 30 °C in an orbital shaker. Glass beads (0.45 mm diameter) filling half the sample volume were added and spheroplasts were broken in ice by performing four rounds of 1 min vortex shakings with 2 min pauses between them. All steps were performed at 4 °C. Cell debris were eliminated by centrifugation at 1000× *g* for 5 min and the supernatant was further centrifuged at 12,000× *g* for 5 min. The enzyme activity was measured in the supernatant as indicated in the commercial kit leaflet and the results were normalized regarding the protein content (expressed in mg) as measured by the Lowry method modified by Markwell [37,38]. 

## 3. Results

### 3.1. Growth in BaP as Carbon Source

We previously assessed the effect of several PAHs on the growth of four different yeast strains (*C. albicans*, *R. mucilaginosa*, *D. hansenii,* and *S. cerevisiae*) in the presence of naphthalene, biphenyl, BaP, and benzene (a non-PAH used as control) [39] (Appendix A). *R. mucilaginosa* growth decreased in the presence naphthalene, biphenyl, and benzene at concentrations above 100 ppm, but BaP caused no effect. If we consider optical density equivalent to biomass, *S. cerevisiae* showed the lowest cell density when exposed to these contaminants. However, although the growth of *C. albicans* was not impaired, this yeast is not suitable for mycoremediation purposes because it is considered an opportunistic pathogen. On the other hand, the growth of *D. hansenii* was not inhibited and it displayed the highest BaP degradation rate (as shown in Table 2). Thus, *D. hansenii* was selected to conduct further experiments using BaP as a xenobiotic. Additionally, this yeast is of interest because of its ability to grow at low temperatures. Furthermore, it is an oleaginous organism that metabolizes different carbon sources, among other relevant physiological and metabolic features [26,27,28].

To investigate the combined effect of temperature and BaP concentration on *D. hansenii* growth, drop tests were performed in the presence of BaP concentrations between 10 and 500 ppm. Plates were incubated at 15, 28, and 37 °C for 72 h, and at 4 °C for one month. Figure 3 shows that *D. hansenii* grew reasonably well at 15 and 28 °C in the presence of BaP as a carbon source and using the nitrogen and carbon provided by the amino acids supplied in the culture medium. Growth was inhibited under all conditions at 37 °C (not shown). After one month at 4 °C, *D. hansenii* grew satisfactorily. Surprisingly, it can also grow using acetone (YNBAc) as a carbon source (included as a control because it was used to solubilize BaP).

To calculate the growth kinetics constant values in the presence of BaP at different temperatures, growth curves were obtained using a liquid YNB medium supplemented with 0 to 500 ppm BaP at 15, 28, and 37 °C. Figure 4a shows that *D. hansenii* grew at 15 °C in the presence of 10–500 ppm BaP. The specific growth rates were very similar at all concentrations, whereas doubling times ranged between 9 and 13 h. The 28 °C temperature is considered optimal for *D. hansenii* growth (Figure 4b) [26,28,40,41], and it was the condition where growth rate was the fastest in the presence of 10 ppm BaP, but it also showed the slowest rate when the concentration was 200 ppm. However, as shown in Figure 4b, the variable effect caused by different treatments was evident as the specific growth rate decreased by increasing BaP concentration. No growth was observed at 37 °C. The table in Figure 4c shows the doubling time and the specific growth rate values obtained in the presence of every BaP concentration and temperature. The results are graphically depicted in Figure 4d. Interestingly, there was slight growth in the YNB medium at two of the temperatures evaluated in the absence of a carbon source. This suggests that *D. hansenii* metabolizes the amino acids supplemented in this medium to provide nitrogen to sustain growth.

It is important to point out that, even in the presence of BaP 500 ppb, *D. hansenii*, increased its specific growth rate at both temperatures. To mathematically obtain this data, we used the logarithmic growth phase values without considering the lag phase data, as it is very extended when compared to all other growth conditions. This indicates that *D. hansenii* slowly adapts to this BaP concentration, but growth was acceptable once it achieved such adaptation. If absorbance is considered equivalent to biomass, the value obtained in the presence of 500 ppm BaP was inferior when compared to all the other growth conditions.

It is noteworthy to mention that the initial OD_600_ setting for the cell suspension was performed in a spectrophotometer and growth monitoring was carried out in a plate reader, therefore, there is a difference regarding the initial OD_600_ value (for extra information refer to Section 2.2). Additionally, all absorbance values measured in the presence of 500 ppm BaP are initially higher because the YNB medium becomes cloudy in the presence of this highest concentration level of BaP and acetone. The suspension becomes clear upon shaking the media and when temperature is constant.

Nonetheless, *D. hansenii* grew in the presence of BaP and its viability was monitored for 10 days (Figure 5). We observed that cell viability was not impaired after exposing the yeast to the pollutant for 10 days. Interestingly, right from day 5, viability decreased by 25% in the YNB medium, probably because of nitrogen and carbon source depletion.

### 3.2. BaP Degradation and Selective Inhibition of Cytochrome P450 (CYP) by Piperonyl Butoxide (PBO)

After a 10-day incubation performed to promote BaP degradation, the remaining amount was quantified by fluorescence (Table 2). To confirm such degradation, culture media extracts were analyzed by GC-MS. BaP retention time was 27 min, as shown in the chromatogram of a BaP standard (130 ppm). The chromatograms obtained after analyzing the extracts supplemented with BaP either in the absence or presence of an inhibitor (piperonyl butoxide, PBO) were compared with that of a yeast-free medium. Several changes were identified regarding the chromatogram pattern, including the amplitude and size of the peaks. This suggests a degradation of BaP, along with the appearance of other unidentified metabolites (Appendix A). 

Importantly, the BaP recovery rate was 90% in a culture medium without cells containing 100 ppm BaP, whereas such recovery was 87% in a medium inoculated with dead yeast cells (killed control).

To confirm a CYP-mediated BaP degradation [18,23], we used a specific CYP inhibitor: PBO [21,23,42,43]. Overnight cultures of *C. albicans*, *D. hansenii*, *R. mucilaginosa,* and *S. cerevisiae* supplemented with 1, 10, and 15 mM PBO were prepared in a YNBG medium. Subsequently, 10-fold serial dilutions were placed as drops in a YNBG medium. It was observed that 10 mM PBO was a suitable concentration to perform the inhibition tests, as it did not inhibit growth (not shown). 

Table 2 shows the BaP degradation percentage either in the absence or presence of PBO. When this CYP inhibitor was not added, *D. hansenii* metabolizes 84% (GC-MS 80%) of the BaP supplemented in the medium. This value was the highest degradation rate. In contrast, *R. mucilaginosa* metabolizes approximately 70% (GC-MS 73 %) and it represents the lowest rate. The presence of a CYP inhibitor (PBO) effectively prevented BaP degradation by all yeasts. This inhibition was less effective in *R. mucilaginosa* as BaP degradation was approximately 15% (GC-MS 17%). For the other three yeasts, BaP degradation was approximately 10% in the presence of PBO. This demonstrates that BaP degradation is mediated by a CYP enzyme in all these yeasts (Table 2).

### 3.3. Genetic and Expression Analysis of the CYP Enzymes Linked to BaP Metabolism

Depending on their ecological niche, some organisms metabolize PAHs through different pathways as part of an adaptation process, e.g., by expressing CYP and other proteins involved in PAH metabolism. This was reported for some filamentous fungi [13,17,18,21].

The *P. chrysosporium AY515589* gene coding for the CYP5144A7 protein was previously studied. It was originally identified in a microarray performed after exposing this fungus to different PAHs (including BaP), and it selectively oxidizes BaP. This gene contains eight introns, and the protein is comprised by 517 amino acid residues [23]. We also considered the sequence of the *AN1884* gene that codes for the *A. nidulans* CYP617D1 protein expressed in the presence of BaP [18].

Takin into account these sequences, we performed a query in several databases (see Section 2) for the existence of homologous genes encoding CYP monooxygenases in all yeasts studied. 

The *C. albicans ALK1* gene (1581 bp) was identified as encoding the CYP52A24 protein (526 amino acid residues). *D. hansenii* harbors the *DEHA2C02596g* gene (1575 bp) that codes for the CYP52A44 protein (525 amino acid residues). *S. cerevisiae* expresses the homologous CYP56A1 protein (489 amino acid residues) that is encoded by the *DIT2* gene (1470 bp). Three possible homologous genes were identified for *R. mucilaginosa*: *g700567* (2497 bp), *g710473* (2373 bp), and *g680755* (2809 bp). These were not previously reported nor characterized. Appendix A summarizes the data for these genes and proteins.

An alignment of the respective amino acid sequences was performed using Clustal Omega (https://www.ebi.ac.uk/Tools/msa/clustalo/, accessed on 22 June 2020) [33]. We identified four conserved domains exclusive of the CYP proteins, as shown in Figure 6a: (i) AGXDTT, an oxygen-binding and activation domain; (ii) and (iii) the EXXR and PER motifs, comprising the E-R-R triad, a structure that forms the heme pocket that also provides stabilization of the core structure; and (iv) FXXGXRXCXG, the heme-binding domain containing an invariant Cys ligand [25,44]. Based on the presence of such motifs, all these proteins may be classified as CYP. Figure 6b depicts the CYP conserved motifs. The presence of these motifs was previously reported for the other taxonomic groups [25,44,45,46].

The protein that metabolizes PAHs and other xenobiotics in *C. albicans* bears a high percentage of identity to that expressed by *D. hansenii* (61.7%). The lowest percentage of identity observed was between the *R. mucilaginosa* homologue and that of *A. nidulans* (19.1%). The identity extent among all the other proteins ranges between 20 and 30%.

### 3.4. Plasmid Construction and the Expression of DhDIT2 in S. cerevisiae

To ensure the expression of a functional CYP protein in *S. cerevisiae*, it was important to analyze the insertion of the *DhDIT2* gene into the pYES2 plasmid. This verification is performed because the *D. hansenii* CTG codon codes for serine instead of leucine [34]. We observed that the *DhDIT2* gene has no CTG codons, thus DNA manipulation was performed using the conventional techniques, as indicated in Figure 2.

A YNB medium containing 2% galactose without uracil was used to verify the correct transformation of the strains. Both the wild-type *S. cerevisiae* BY4742 and its *dit2Δ* mutant fail to grow in this medium because they are uracil auxotrophic. The *DhDIT2* gene was inserted under the control of the galactose promoter contained in the pYES2 plasmid. The transformed strains were able to grow in this medium, as such plasmid contains the *URA* gene (Figure 7a).

Figure 7b shows the effect of 100 ppm BaP on both wild-type and transformant growth. The gene *ScDIT2* is required to metabolize BaP, as no considerable growth was observed for the *S. cerevisiae dit2Δ* mutant. Conversely, the *dit2Δ* strain expressing the *DhDIT2* gene grew similarly as a wild-type strain that also expresses the *DhDIT2* gene. These two strains displayed an optimal growth when compared to the *S. cerevisiae* BY4742 strain with no plasmid and to *D. hansenii*.

The expression of *DhDIT2* in the BY4742 wild type improved its growth in the presence of BaP. Thus, it was evaluated if the growth of a *S. cerevisiae* strain expressing the *D. hansenii DIT2* gene was affected by a lower temperature (15 °C). A drop test was performed using the following media: YNB, YNB supplemented with acetone, YNBG, YNB containing 0.2% galactose and 100 ppm BaP, and YNB supplemented with 100 ppm BaP. All cultures were incubated for 72 h. Appendix A shows that the *S. cerevisiae* pYES2-*DhDIT2* strain displayed a deficient growth in the YNB medium. However, this strain grew optimally when it was cultured in a medium supplemented with either acetone, galactose with BaP, or BaP alone. This confirms its ability to grow in the presence of 100 ppm BaP at 15 °C.

### 3.5. BaP Degradation after 24 h Using a Higher Amount of Biomass

A brief 24 h degradation time was used to evaluate the ability of the wild-type and transformed yeasts to metabolize BaP. The experiment was conducted using a higher amount of yeast biomass (1 g wet weight) previously grown for 24 h in YNB medium supplemented with either glucose or galactose. The preculture was placed in 20 mL of YNB medium containing 0.2% galactose (for those strains transformed with the pYES2 plasmid) supplemented with 100 ppm BaP (OD_600_ = 5.5). Degradation was allowed for 24 h at 28 °C. Afterwards, the remaining BaP was extracted and quantified as described in the Materials and Methods, Section 2.4.

Table 3 shows the percentage of BaP degradation after 24 h using 1 g of yeast biomass. Wild-type yeasts metabolized approximately 50–60% of BaP. *D. hansenii* exhibited an extensive degradation (58.5%, GC-MS 56%). The contribution of the *ScDIT2* gene to BaP degradation was demonstrated as its metabolism decreased to 9.8% (GC-MS 13%) in the absence of the gene. The heterologous *DhDIT2* expression in a *S. cerevisiae dit2Δ* mutant restored the BaP-metabolizing phenotype. This confirms that this *D. hansenii* gene is a central component involved in xenobiotic metabolism. Furthermore, the expression of *DhDIT2* in the *S. cerevisiae* BY4742 wild type also improved BaP degradation.

### 3.6. NADPH-Cytochrome C Reductase Activity after BaP Exposure

It is technically challenging to directly assess the CYP enzyme activity. However, a functional assay may be established as CYP is located in the endoplasmic reticulum, and it may be coupled to the NADPH-cytochrome P450 reductase (CPR) activity that acts as an electron donor protein to activate molecular oxygen. Thus, CYP activity was measured by monitoring CPR [25,47]. It is important to mention that a value of 1 was defined for the enzyme activity on day 0 as each strain displayed different basal values. Subsequently, enzyme activity was reported each day as a fold-increased value regarding the day 0 control.

The strains that optimally grew in the presence of BaP exhibited higher enzyme activities when compared to those grown in glucose (Figure 8). This fact further confirms a CYP- and CPR-mediated BaP oxidation. All strains (both wild-type and transformants) showed a peak enzyme activity on day 2, and such value drastically decreased by day 6.

Among the wild-type strains, *D. hansenii* exhibited the highest enzyme activity value at day 2 (51.2 mU/mg protein), whereas the activity was 46.8 mU/mg protein for *S. cerevisiae* BY4742. The *S. cerevisiae dit2Δ* strain displayed very low activity (4.5 mU/mg protein), and that of the pYES2-*DhDIT2*-transformed *dit2Δ* strain was 24.2 mU/mg protein in the presence of BaP (day 3). This again demonstrates the relevance of the *ScDIT2* gene for BaP metabolism. CPR enzyme activity increased 24-fold (60.3 mU/mg protein) on day 2 when the *DhDIT2* gene was expressed in the *S. cerevisiae* BY4742 strain.

## 4. Discussion

Yeasts thrive in a wide variety of environments: contaminated soils, cold and hot deserts, glacier ice and water, hyper saline lakes, and deep waters [20,48]. Therefore, it is not surprising that they possess enzyme systems that enable them to degrade and/or tolerate xenobiotics, such as PAHs.

Some yeasts were isolated from sites contaminated by PAHs and their ability to degrade the pollutants was measured in vitro. These yeasts included *D. hansenii*, *C. albicans*, *R. mucilaginosa,* and *S. cerevisiae* [2,13,15,49,50,51,52]. The genomes of taxonomically diverse yeast species were compared, and they were identified as potential xenobiotics metabolizers (including PAHs) [20].

In actual environmental conditions, bioremediation is only efficient in sites where climatic conditions support microbial growth and performance. Adequate temperatures are also important to enable the expression of the PAH-degrading enzymes [2]. It is known that *D. hansenii* grow at low temperatures [53], but when BaP is used as a carbon source, such growth is impaired. Nevertheless, this yeast grows within a 30-day period at 4 °C. This probably occurs because BaP solubility decreases at low temperatures and its bioavailability is affected [2,48]. BaP solubility increases at 37 °C, but this temperature is restrictive for *D. hansenii* [26,40]. Interestingly, we detected that *D. hansenii* grow in a YNB medium containing acetone, the solvent used to solubilize BaP. This was previously reported for *S. cerevisiae* as it converts acetone to acetoacetyl-CoA [54]. 

The BaP degradation rate values we observed for all of yeasts agree with those already reported for other organisms. This indicates that they all metabolize these molecules and growth occurs in despite their presence [15,21,49,55,56,57,58,59,60].

Yadav and Loper in 1999 [61] pointed out that CYP is relevant to achieve the assimilation and degradation of both PAHs and alkanes, whereas Smith et al., in 2004 [62] confirmed that these compounds may be used as carbon sources. Additionally, Mandal and Das in 2016 and 2018 [57,63,64] reported that two different yeast consortiums, YC02 (integrated by *Hanseniaspora opuntiae* NS02, *D. hansenii* NS03, and *Hanseniaspora valbyensis* NS04) and YC04 (*Rhodotorula* sp NS01, *D. hansenii* NS03, and *Hanseniaspora valbyensis* NS04) metabolize perylene and benzo[ghi]perylene, respectively.

*D. hansenii* is a yeast commonly found in marine environments because of its tolerance to high salt levels and its ability to grow in a wide range of carbon sources [65,66]. Moreover, it is well established that this yeast metabolizes the contaminants occurring in fresh and salty waters [45,67,68,69]. 

Based on our results obtained during the BaP degradation tests using four different yeasts (*C. albicans*, *D. hansenii*, *R. mucilaginosa,* and *S. cerevisiae*) and considering that BaP may be metabolized by a CYP enzyme in *A. nidulans* [18], we conducted our experiments in the presence of a specific inhibitor (PBO). PBO did not inhibit the growth of any of the four yeasts we studied [42,43]. As CYP does not participate in the primary metabolism reactions [25,70], it was proposed that its main function is to enable adaptation to allow survival in ecological niches [25]. PBO blocks the porphyrin iron contained in the heme group of the CYP enzyme complex [23,42,43]. The results observed regarding the BaP degradation by our four studied yeasts in the presence of PBO are comparable to those obtained for *Aspergillus sydowii*: only 10% of BaP was metabolized when 10 mM PBO was included [21].

A low similarity among CYPs was previously reported, as their classification is solely based on their amino acid sequence, i.e., those with a similarity above 40% comprise a single CYP family, and those above a 55% similarity represent subfamilies. Despite their wide diversity and low sequence similarities, CYP enzymes possess four distinctive motifs that, along with their conserved tertiary structures, support their enzymatic function [25,44]. 

CYP proteins are induced by xenobiotics [71]. To test the relevance of the *ScDIT2* gene, we performed a complementation strategy using a *S. cerevisiae dit2Δ* mutant with a *DhDIT2* gene. Additionally, to obtain a strain expressing an extra CYP, regardless of that encoded by the *ScDIT2* gene, we transformed the wild-type strain *S. cerevisiae* BY4742 with the *DhDIT2* gene, to test if there was an improvement in the BaP degradation capacity. 

The *ScDIT2* gene encodes the CYP protein, that is responsible for metabolizing BaP and other PAHs during the phase I reactions. It also participates in the ergosterol biosynthesis pathway and during sporulation [70]. Our results demonstrate its role in the degradation of BaP because the insufficient growth of the *S. cerevisiae dit2Δ* mutant was restored in a transformant harboring the *DhDIT2* gene. Two CYP proteins were previously identified in *D. hansenii.* These were classified as alkane hydroxylases as they oxidize *n*-alkanes and they were termed CYP52A12 and CYP52A13 (encoded by the *DH-ALK1* and *DH-ALK2* genes, respectively). The similarity between them is 60% and they bear a 61% identity regarding the *Candida* sp. CYP52A3 protein (encoded by the *ALK1* gene) [61]. An exhaustive query was conducted on the NCBI database for either the CYP52A12 (UniProtKB/Swiss-Prot: Q9Y757.2) or *DH-ALK1* (GenBank: AAD22536.2) sequences, and either the CYP52A13 (UniProtKB/Swiss-Prot: Q9Y758.1) or *DH-ALK2* sequences (GenBank: AAD22537.1). We found that they are two different CYP proteins when compared to that encoded by the *DhDIT2* gene. A MycoCosm comparison between the CYP52A12 and CYP52A13 sequences and the protein encoded by *DhDIT2* revealed a 63.8 and 65.9% identity, respectively [61,72,73]. This demonstrates that *DhDIT2* codes for a CYP protein distinct to those previously reported.

CYP’s heterologous expression was previously reported using recombinant plasmids. Yang et al. in 2008 [74] expressed the *Helicoverpa armigera* CYP9A12 and CYP9A14, proteins in *S. cerevisiae* and this resulted in their ability to demethylate pyrethroids. Moreover, Syed et al. in 2010 [23] cloned the cDNA of different *P. chrysosporium* CYPs into a plasmid and then transformed *Pichia pastoris*, conferring this yeast with the ability to metabolize multiple PAHs, including BaP. These and other studies support the construction and transformation strategies that we performed using *S. cerevisiae* strains. Our results demonstrate that the *ScDIT2* gene is vital for BaP metabolism because the *dit2Δ* mutant complemented with the *DhDIT2* gene recovered the BaP-metabolizing phenotype. On the other hand, the heterologous expression of *DhDIT2* in the wild-type strain resulted in an enhanced BaP degradation rate and conferred it with the ability to grow in BaP at low temperatures (15 °C), similarly to the results observed for *D. hansenii*.

CPR is a membrane-bound electron donor diflavin protein essential for CYP-mediated reactions, as it sequentially delivers two electrons to activate molecular oxygen. Thus, CYP activity may be indirectly measured using the NADPH-cytochrome P450 reductase assay, as there are several factors (e.g., air) that interfere with a direct detection [25,47]. Interestingly, the results show a day where the enzyme activity peaked, similarly to that observed in *R. mucilaginosa* [20]. The heterologous expression of CYP increased the CPR enzyme activity, as shown by comparing the *S. cerevisiae* BY4742 pYES2-*DhDIT2* with its wild-type strain.

Our results also show that BaP degradation is mediated by enzymes from the CYP superfamily, as demonstrated by an increase in CPR activity and the effects observed when a specific CYP inhibitor (PBO) was used. Furthermore, the heterologous expression of the *DhDIT2* gene induced a short-term enhanced degradation rate in *S. cerevisiae*. Taking this into account, we propose that *D. hansenii* and the *S. cerevisiae* BY4742 pYES2-*DhDIT2* strain may be used for mycoremediation purposes, as they are able to convert PAHs into harmless substances [75]. Both strains were able to tolerate BaP at temperatures between 15 °C and 28 °C, thus they are suitable for contaminated sites within this temperature range. 

The relative expression of the genes participating in all three phases of BaP metabolism (including other PAHs) will be further studied to understand their occurrence over time and to elucidate the molecular pathways involved in the adaptation of *D. hansenii* and *S. cerevisiae* to metabolize BaP.

## 5. Conclusions

We identified the ability of *D. hansenii* to metabolize BaP through a CYP-mediated pathway. Our results show that its growth depends on factors such as temperature and xenobiotic concentration when exposed to BaP. The *ScDIT2* gene is essential for BaP metabolism in *S. cerevisiae,* and the heterologous expression of the *DhDIT2* gene in a *S. cerevisiae* mutant restored its ability to metabolize BaP. As both *D. hansenii* and the *S. cerevisiae* BY4742 pYES2-*DhDIT2* strain grow in the presence of BaP at infra-optimal temperatures, they are proposed as suitable candidates for bioremediation purposes in sites contaminated with BaP. *D. hansenii* was originally isolated from marine sources, and its ability to adapt to these environments is an advantage that must be harnessed to eliminate BaP in marine sites.

## Figures and Tables

**Figure 1 jof-08-01150-f001:**
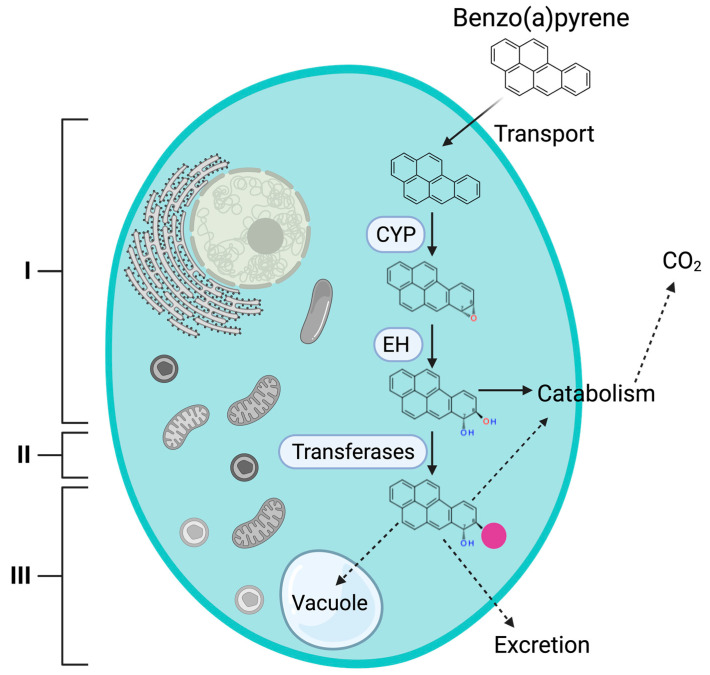
BaP detoxification and degradation system in yeast and other Ascomycota fungi. Phase I: BaP is transported into the cell where it is metabolized by enzymes, such as the cytochrome P450 monooxygenase (CYP) and epoxide hydrolase (EH). Phase II: Transferases (glycosyl-, sulfo-, and glutathione-S-transferases form conjugated molecules using the hydroxylated molecule as substrate (the pink circle represents the sulphate, xylose, glucose, or glutathione moiety that renders them less toxic and more soluble). Phase III: Internalization into the vacuole or excretion of the conjugated molecule and subsequent assimilation by other organisms [13]. Created with BioRender.com.

**Figure 2 jof-08-01150-f002:**
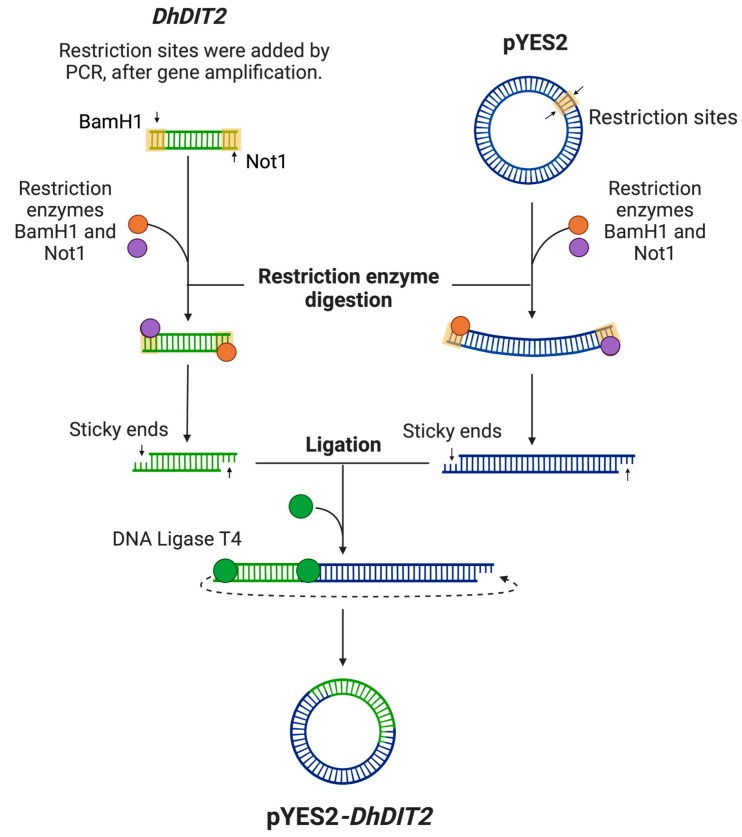
Construction of the pYES2*-DhDIT2* plasmid. Experimental strategy to construct the plasmid expressing the *DhDIT2* gene used to transform the *S. cerevisiae* BY4742 parent strain and the *dit2Δ* mutant. Created with BioRender.com.

**Figure 3 jof-08-01150-f003:**
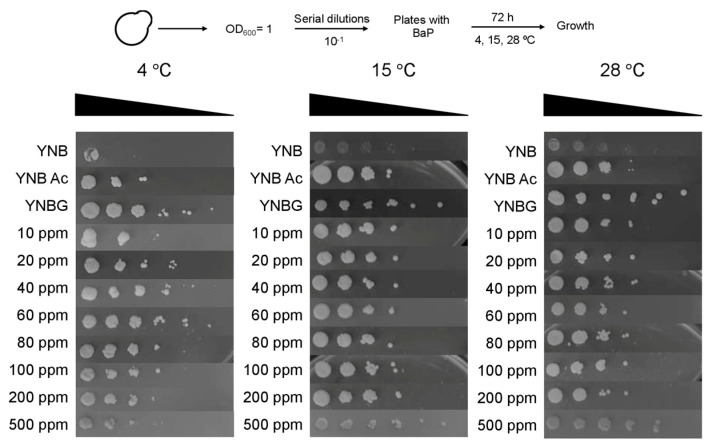
Effect of different benzo(a)pyrene (BaP) concentrations and temperatures on *D. hansenii* growth. Drop tests were performed in YNB medium using BaP as a carbon source. Ten−fold serial dilutions from a yeast suspension adjusted to OD_600_ = 1.0 were dropped in plates containing different BaP concentrations as indicated. They were incubated at 4 °C for one month and at 15 and 28 °C for 72 h. Controls: YNB, yeast nitrogen base; YNBG, YNB supplemented with glucose; YNB Ac, YNB containing acetone (the solvent used to solubilize BaP in a volume equivalent to that of the highest concentration). Representative images of three independent experiments are shown.

**Figure 4 jof-08-01150-f004:**
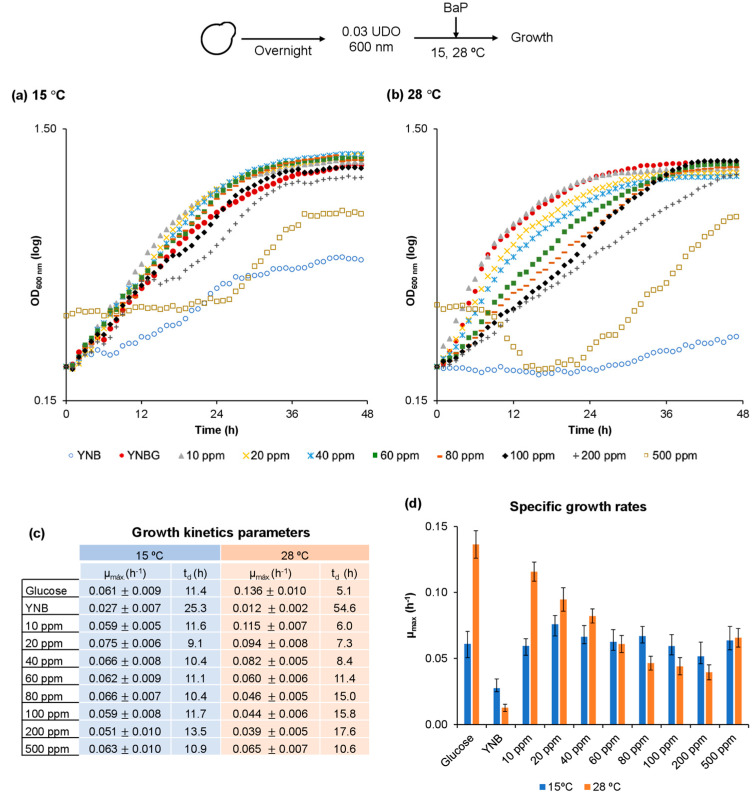
The effect of benzo(a)pyrene and temperature on *D. hansenii* growth. The effect of different BaP concentrations on *D. hansenii* growth was studied at two different temperatures: (**a**) 15 °C, (**b**) 28 °C to obtain (**c**) growth kinetics parameters, and (**d**) specific growth rates at different BaP concentrations and temperatures. The curves depict a representative result of three experimental replicates, each one performed by triplicate. The data in (**c**,**d**) represent the average values ± SD (standard deviation) of the nine results obtained from the curves.

**Figure 5 jof-08-01150-f005:**
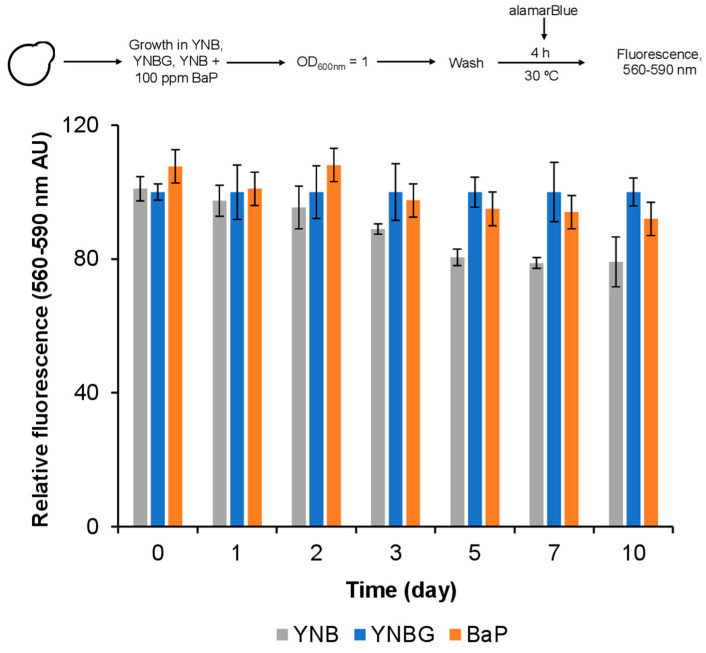
Long-term viability of *D. hansenii* in the presence of 100 ppm BaP. Cells were grown in YNB or YNB supplemented with either glucose or 100 ppm BaP during 1 to 10 days. After the respective time, an aliquot was used to assess cell viability using an alamarBlue^TM^ kit, as indicated in the Section 2. The fluorescence reading of those cells grown in YNB containing glucose was established as the 100% viability value. Results are shown as average value ± SD of three independent experiments performed by quadruplicate.

**Figure 6 jof-08-01150-f006:**
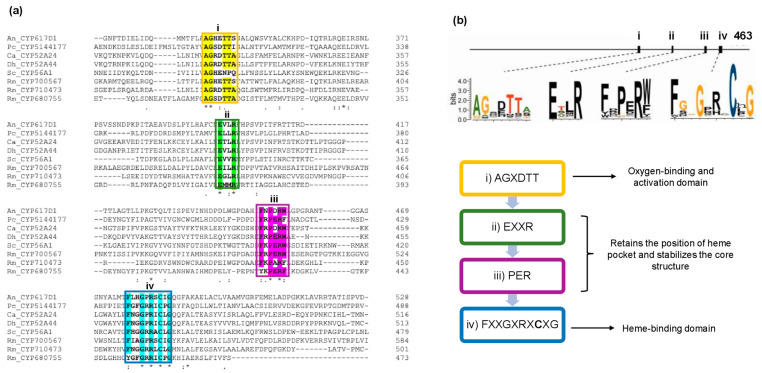
Molecular alignment of the CYP proteins and their conserved motifs. (**a**) Sequence alignment of the CYP proteins identified in different fungal and yeast species: An, *Aspergillus nidulans*; Pc, *Phanerochaete chrysosporium*; Ca: *Candida albicans*; Dh, *Debaryomyces hansenii;* Rm, *Rhodotorula mucilaginosa* and Sc, *Saccharomyces cerevisiae*. The conserved domains labelled as i–iv are shown inside the colored boxes and they are explained in detail in the text. The Clustal Omega online tool was used to perform the analysis (https://www.ebi.ac.uk/Tools/msa/clustalo/ accessed on 22 June 2020) [33]. The positions containing a single and fully conserved residue are marked with (*); the conserved groups with highly similar properties are marked with (:); the conserved groups with properties bearing weak similarity are marked with (.) [33]. (**b**) Conserved sequences of the eukaryotic CYP motifs [25,44]. The boxes indicate the function of the respective motifs.

**Figure 7 jof-08-01150-f007:**
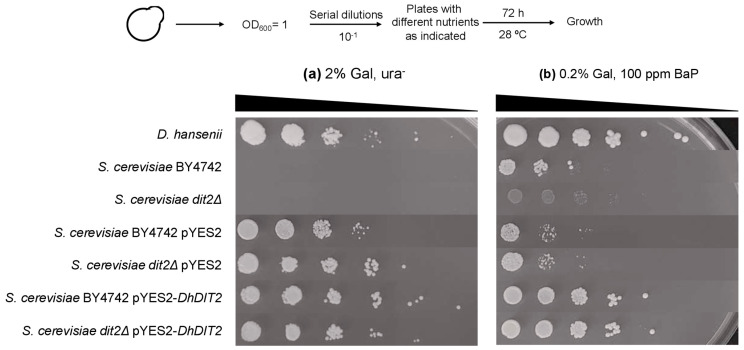
Effect of BaP on the growth shown by the wild-type *S. cerevisiae* and *D. hansenii*, as well as the *S. cerevisiae dit2Δ* mutant transformed with the *DhDIT2* gene. The indicated strains were grown overnight in either liquid YNBG or YNBGal media and the cultures were adjusted to OD_600_ = 1.0. Aliquots of 10-fold serial dilutions were spotted in YNB medium containing 2% galactose without uracil (**a**) or in YNB medium containing 0.2% galactose supplemented with 100 ppm BaP (**b**). The plates were incubated at 28 °C for 72 h and photographed afterwards. A representative image of three independent experiments is shown.

**Figure 8 jof-08-01150-f008:**
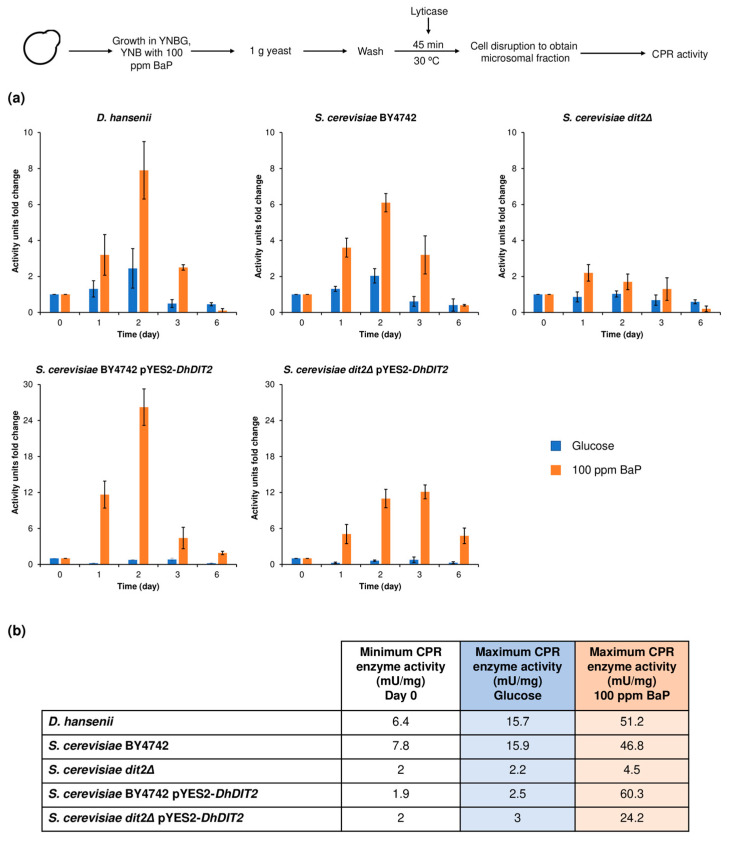
NADPH-cytochrome *c* reductase (CPR) activity during a 6-day exposure to BaP. (**a**) Increase in CPR enzyme activity (fold–change). (**b**) Maximum and minimum CPR activity. Cells were grown for 24 h in YNB medium supplemented either with 2% glucose or 2% galactose. Then, they were incubated in YNB medium containing either 0.2% glucose or 0.2% galactose supplemented with 100 ppm BaP. Each day, the microsomal fraction was extracted using 1 g (wet weight) of biomass and CPR enzyme activity was measured as indicated in the Section 2. Results represent an average value ± SD of three independent experiments.

**Table 1 jof-08-01150-t001:** Primers used for plasmid construction.

Primer	Sequence
dhdita	AATATGTCTACCGATAAATTAAAATCTTATGTTGAAGAAATATC
dhditb	TATCTACTTATTTAAGGAAATAAAAACACCATCCTGATG
dh450a	CATGGATCCAATATGTCTACCGATAAA
dh450c	GCGGCCGCTATCTACTTATTTAAGGA

**Table 2 jof-08-01150-t002:** The degradation of benzo(a)pyrene (BaP) and its inhibition induced by piperonyl butoxide (PBO).

	% BaP Degradation without PBO *	% BaP Degradation with PBO *
*C. albicans*	77.0 ± 3.9	9.7 ± 2.5
*D. hansenii*	84.1 ± 2.9	9.5 ± 1.9
*R. mucilaginosa*	70.0 ± 4.9	14.9 ± 1.0
*S. cerevisiae*	79.5 ± 2.9	9.7 ± 1.9

Yeasts were grown in a YNBG medium for 24 h and subsequently transferred to a YNB medium supplemented with 100 ppm BaP, either in the absence or presence of 10 mM PBO. After a 10-day incubation, an aliquot was collected, and the remaining BaP was quantified by fluorescence as described in the Section 2. Results are reported as average values ± SD obtained with 3 to 5 samples. * Confirmatory GC-MS results: without inhibitor: *C. albicans* 75%, *D. hansenii* 80%, *R. mucilaginosa* 73%, and *S. cerevisiae* 76%; in the presence of the inhibitor: *C. albicans* 12%, *D. hansenii* 15%, *R. mucilaginosa* 17%, and *S. cerevisiae* 9%.

**Table 3 jof-08-01150-t003:** Percentage of BaP degradation by different yeasts after 24 h.

Strain	% BaP Degradation *
*D. hansenii*	58.5 ± 2.7
*S. cerevisiae* BY4742	51.3 ± 4.2
*S. cerevisiae dit2Δ*	9.8 ± 2.0
*S. cerevisiae* BY4742 pYES2*-DhDIT2*	66.4 ± 2.58
*S. cerevisiae dit2Δ* pYES2*-DhDIT2*	46.0 ± 3.51

Yeasts were grown for 24 h either in YNBG or YNB-Gal (for those strains harboring the pYES2 plasmid) media. Cells were harvested by centrifugation, and they were submitted to fasting in water for 24 h. The ability to metabolize BaP was evaluated for 24 h in 20 mL of YNB media containing 0.2% galactose and 100 ppm BaP, using 1 g of biomass (wet weight; OD_600_ = 5.5) at 28 °C and constant shaking at 250 rpm. After the incubation, the remaining BaP was extracted, and it was quantified by fluorescence as described in the Section 2. Results are expressed as average values ± SD from three independent experiments. * Confirmatory GC-MS results: *D. hansenii* 56%, *S. cerevisiae* BY4742 48%*, S. cerevisiae dit2Δ* 13%*, S. cerevisiae* BY4742 pYES2*-DhDIT2* 61%*,* and *S. cerevisiae dit2Δ* pYES2*-DhDIT2* 41%.

## Data Availability

Not applicable.

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
