# Peer review of "DhDIT2* Encodes a *Debaryomyces hansenii* Cytochrome P450 Involved in Benzo(a)pyrene Degradation—A Proposal for Mycoremediation"

_jof, 2022, doi:10.3390/jof8111150_

Round 1

Reviewer 1 Report

Authors provide an interesting study about several new genes capable of degradation of benzo(a)pyrene, the degradation, and heterologous expression experiments were conducted to verify the ability of DhDIT2. the study is meaningful for bioremediation of PAHs in the environments, but not quite complete. If possible, I recommend that the degradation products and pathways should be added, which is important for us to understand the fate of PAHs in environments and transform them into harmless products. A major revision would be appropriate. 

Here are some comments that should be concerned. 

1.     The scientific language needs to be improved and the contents of manuscript should be checked again. (e.g., line 18 “in the microorganisms studied”; line 49, “Dehnen, et al.,”; line 144, “DO600=0.1” and so on)

2.     In the part of “Introduction”, some cited data about the degree of BaP pollution should be supplemented. And paragraphs with similar content should be integrated, there are too many paragraphs.

3.     In the degradation experiments, whether spontaneous degradation of BaP in the YNB/YNBG/YNBAc medium occurs? What is the value of spontaneous degradation, respectively? And in part of 2.4, what is the recovery rate of BaP?

4.     Line 155, the author should add more detailed information on transformation (transformation method, how to prepare competence cell, et al.)

5.     In Fig. 2a and b, can the author explain why the start point of 500 ppm group is different from the other groups? It seems that it (500 ppm group) has a higher cell density at the beginning.

Small issues:

1.     At the top of Fig 2, what does “0.03 UDO 600nm” mean?

2.     Here are no fig. 6a/b in fig 6, you should check this.

3.     Line 349, what is the exact cell density when 1 g yeast was added to the medium?

4.     In the YNB/YNBGG/YNBAc medium, amino acids is not only N sources but also C sources, so some statements in growth experiments (such as line 196-197) should be corrected.

5.     Line 15 and 420, format error (space issue)

Author Response

Reviewer 1:

Review Report Form

Open Review

English language and style

( )Extensive editing of English language and style required
(x) Moderate English changes required
( ) English language and style are fine/minor spell check required
( ) I don't feel qualified to judge about the English language and style

Yes

Can be improved

Must be improved

Not applicable

Does the introduction provide sufficient background and include all relevant references?

( )

( )

(x)

( )

Are all the cited references relevant to the research?

( )

(x)

( )

( )

Is the research design appropriate?

( )

(x)

( )

( )

Are the methods adequately described?

( )

( )

(x)

( )

Are the results clearly presented?

( )

(x)

( )

( )

Are the conclusions supported by the results?

( )

(x)

( )

( )

Comments and Suggestions for Authors:

Authors provide an interesting study about several new genes capable of degradation of benzo(a)pyrene, the degradation, and heterologous expression experiments were conducted to verify the ability of DhDIT2. The study is meaningful for bioremediation of PAHs in the environments, but not quite complete. If possible, I recommend that the degradation products and pathways should be added, which is important for us to understand the fate of PAHs in environments and transform them into harmless products. A major revision would be appropriate. 

Response: Thank you very much for pointing out the lack of information in the introduction of a metabolic pathway that allows the reader to be introduced to understanding the basis of our work. We have now added figure 1 and a more detailed explanation with your suggestion.

Here are some comments that should be concerned. 

  1. The scientific language needs to be improved and the contents of manuscript should be checked again. (e.g., line 18 “in the microorganisms studied; line 49, “Dehnen, et al.; line 144, “DO600=0.1” and so on)

Response to point 1. We apologize for this. Being not native English speakers is always hard. We are sending now our manuscript reviewed by a proofreading certificated company that made the appropriate edition.

  1. In the part of “Introduction”, some cited data about the degree of BaP pollution should be supplemented. And paragraphs with similar content should be integrated, there are too many paragraphs.

Response to point 2. We have now included in the Introduction more information about the pollution caused by BaP with the appropriate bibliography hoping to fulfill your suggestion. Definitively, this information was missing. We also made an adjustment in the redaction so similar information from separate paragraphs is now integrated. Thank you for this useful recommendation.

  1. In the degradation experiments, whether spontaneous degradation of BaP in the YNB/YNBG/YNBAc medium occurs? What is the value of spontaneous degradation, respectively? And in part of 2.4, what is the recovery rate of BaP?

Response to point 3. More experiments were conducted in order to measure BaP by GC-MS from the suggestion of another reviewer, and now, all data in table 2 are confirmed. From these data, we calculated the spontaneous degradation, which is zero, and the recovery rate that corresponds to 90 % (now included in the table description).

  1. Line 155, the author should add more detailed information on transformation (transformation method, how to prepare competence cell, et al.)

Response to point 4. We are really sorry to had written such valuable information in a section to which it did not belong, in the current version, the missing information has been added, now in the Materials and methods section. We appreciate the suggestion and comment.

  1. In Fig. 2a and b, can the author explain why the start point of 500 ppm group is different from the other groups? It seems that it (500 ppm group) has a higher cell density at the beginning.

Response to point 5. This information has already been included in the new version of the article, now figure 3.  You are right, it seemed that the 500 ppm group has a higher cell density, however it was a matter of the low solubility of BaP at the beginning of the experiment. As it is now explained in the text, as the medium reaches 30 °C, and the experiment proceeds, cloudiness is cleared.

Small issues:

  1. At the top of Fig 2, what does “0.03 UDO 600nm” mean? Changed to OD600 (Optical Density measured at 600 nm), we apologize for this mistake.
  2. Here are no fig. 6a/b in fig 6, you should check this. Now it is Figure 7 and has been modified to be 7a and 7b. Thanks for noting this.
  3. Line 349, what is the exact cell density when 1 g yeast was added to the medium? We have now measured this. In the Materials and methods section and in the corresponding table it is now explicitly added that in 20 mL with 1 g (w.w.) of yeast, we have 5.5 OD600 units.
  4. In the YNB/YNBGG/YNBAc medium, amino acids is not only N sources but also C sources, so some statements in growth experiments (such as line 196-197) should be corrected. This is already corrected in the text. Thank you for your suggestion.
  5. Line 15 and 420, format error (space issue) This is now corrected, thank you.

We would like to thank the reviewer for taking the time to do a proper review of our work and thereby improving our paper.

Reviewer 2 Report

Summary: 

The authors aim to understand the effect of PAHs such as benzo(a)pyrene (BaP) on the growth of fungal and yeast species: Candida albicans, Debaryomyces hansenii, Rhodotorula mucilaginosa and Saccharomyces cerevisiae. The paper highlights that the detoxification or mechanisms were mediated by cytochrome P450 (CYP) that was induced in the presence of these xenobiotics. The authors study the effect of various factors such as temperature and contaminant concentrations on the enzyme activity and degradation of BaP. In addition, they studied the genes associated with the metabolism of BaP and observed the isolating and transforming S. cerevisiae, with a gene reported to be involved in the detoxification metabolism from D. hansenii, for degradation potential. 

Comments:

1.     The authors should restructure the abstract part. The logic is not working well and novelty of the work is relatively weak. It must contain the most significant results.

2.     What are the environmentally relevant concentrations of BaP and other PAHs studied in this paper, and was a test at those concentrations also conducted after the data presented in the paper were procured? Might be helpful to use the 0.2 ppb (I think is the EPA approved limit), for the data presented in Table 2.  

3.     Samples were taken at days 0, 1, 2, 3, and 6 (line 169); (lines 114-115) samples taken at 0, 1, 2, 3, 5 and 7 etc. – I think it would be helpful to state the days samples were collected for all the various steps in the protocol, to provide some understanding of the timeline of the degradation activity or growth etc. 

4.     Figure 1 – Image J could perhaps be used to comparatively quantify the growth of D. hansenii perhaps, in addition to the kinetics, to correlate the concentration of BPA with microbes.

5.     In Fig.2(a) and (b), at both temperatures, the final OD all reached the same value eventually and the difference of growth rate was not that obvious. The data doesn’t seem to match the conclusion of cited work.

6.     Fig. 2A: Use different shapes can be used for the concentrations to prevent confusion with colors since red and green are both present in the graph

7.     Only abiotic control is not enough. Killed controls should also be evaluated to eliminate the effects of adsorption by cells.

8.     In Table 3, why with the same heterologous expression of DhDIT2, the BaP degradation by S. cerevisiae BY4742 and S. cerevisiaedit2Δ were different?

9.     In line 114, change DO600 to OD600

10.  Line 186 – Why is the non-PAH control not shown and only a citation provided? 

11.  In Fig. 7(b), wild type D. hansenii and S. cerevisiae BY4742 has similar or even higher CYP activity compared with the ones with DhDIT2 gene expression. Explain why BaP degradation was higher by S. cerevisiae BY4742 and S. cerevisiae dit2Δ-pYES2-DhDIT2? Does that mean the CYP activity is not important in BaP degradation?

12.  Explain why this work evaluated temperature effects on BaP degradation only on wild type D. hansenii? Based on this study, wild type D. hansenii will not be further used in mycoremediation. The temperature effects should be studied on S. cerevisiae BY4742-pYES2-DhDIT2 as well.

13.  Figure 7 is very compelling!

14.  The conclusion section is missing some perspectives related to future research work.

Author Response

REVIEWER 2

Open Review

English language and style

( )Extensive editing of English language and style required
( ) Moderate English changes required
(x) English language and style are fine/minor spell check required
( ) I don't feel qualified to judge about the English language and style

Yes

Can be improved

Must be improved

Not applicable

Does the introduction provide sufficient background and include all relevant references?

( )

(x)

( )

( )

Are all the cited references relevant to the research?

( )

( )

(x)

( )

Is the research design appropriate?

( )

(x)

( )

( )

Are the methods adequately described?

( )

( )

(x)

( )

Are the results clearly presented?

( )

(x)

( )

( )

Are the conclusions supported by the results?

( )

( )

(x)

( )

Comments and Suggestions for Authors

Summary: 

The authors aim to understand the effect of PAHs such as benzo(a)pyrene (BaP) on the growth of fungal and yeast species: Candida albicans, Debaryomyces hansenii, Rhodotorula mucilaginosa and Saccharomyces cerevisiae. The paper highlights that the detoxification or mechanisms were mediated by cytochrome P450 (CYP) that was induced in the presence of these xenobiotics. The authors study the effect of various factors such as temperature and contaminant concentrations on the enzyme activity and degradation of BaP. In addition, they studied the genes associated with the metabolism of BaP and observed the isolating and transforming S. cerevisiae, with a gene reported to be involved in the detoxification metabolism from D. hansenii, for degradation potential. 

Thank you very much for your very insightful summary and commentary on our work.

Comments:

  1. The authors should restructure the abstract part. The logic is not working well and novelty of the work is relatively weak. It must contain the most significant results.

      Response to point 1. We appreciate this appropriate comment. The abstract is now modified as suggested.

  1. What are the environmentally relevant concentrations of BaP and other PAHs studied in this paper, and was a test at those concentrations also conducted after the data presented in the paper were procured? Might be helpful to use the 0.2 ppb (I think is the EPA approved limit), for the data presented in Table 2.  

      Response to point 2. The environmentally relevant concentrations of BaP and other PAHs are now documented at the Introduction section, citing the suitable references. Higher concentrations than the one you are suggesting were used all along the experiments, based in other reports made with fungi and other yeasts, properly cited in our manuscript, and from previous experiments done in our lab, now included in Figure S1.  

  1. Samples were taken at days 0, 1, 2, 3, and 6 (line 169); (lines 114-115) samples taken at 0, 1, 2, 3, 5 and 7 etc. – I think it would be helpful to state the days samples were collected for all the various steps in the protocol, to provide some understanding of the timeline of the degradation activity or growth etc. 

      Response to point 3. According to the degradation curve, these days were selected for the enzyme activity and viability experiments. From day 3 onwards, the degradation curve did not change much, so it was decided to leave only one point at day 6 for the enzyme activity. This information is now also indicated in the text. Thank you for the comment.

  1. Figure 1 – Image J could perhaps be used to comparatively quantify the growth of D. hanseniiperhaps, in addition to the kinetics, to correlate the concentration of BPA with microbes.

      Response to point 4. Thanks for the suggestion, but usually, in reports where the effect of a compound on yeast growth is analyzed, the drop tests with decimal serial dilutions analysis are performed for a qualitative result, as the presented in our manuscript (now Figure 3). According to the literature (Kwolek-Mirek & Zadrag-Tecza, 2014), performing a spotting test provides reliable information about cell viability, which in our case, was the initial objective of that experiment. Further quantitative information is displayed at Figure 4, precisely at the kinetics experiment as you noted.

      Kwolek-Mirek, M., & Zadrag-Tecza, R. (2014). Comparison of methods used for assessing the viability and vitality of yeast cells. FEMS yeast research, 14(7), 1068–1079. https://doi.org/10.1111/1567-1364.12202

  1. In Fig.2 (a) and (b), at both temperatures, the final OD all reached the same value eventually and the difference of growth rate was not that obvious. The data doesn’t seem to match the conclusion of cited work.

      Response to point 5. We are very sorry for the errors you mention, but they have been corrected in the text. Thank you.

  1. Fig. 2A: Use different shapes can be used for the concentrations to prevent confusion with colors since red and green are both present in the graph

                  Response to point 6. Thank you for your comment, the figures have been corrected.

  1. Only abiotic control is not enough. Killed controls should also be evaluated to eliminate the effects of adsorption by cells.

      Response to point 7. This control has already been added and the information is included in the text.

  1. In Table 3, why with the same heterologous expression of DhDIT2, the BaP degradation by S. cerevisiae BY4742 and S. cerevisiae dit2Δ were different?

      Response to point 8. This is because the wild-type strain (S. cerevisiae BY4742) has its own CYP (encoded by ScDIT2), whereas the S. cerevisiae dit2Δ strain lacks the ScDIT2 gene, and both strains when transformed get the DhDIT2 gene.

  1. In line 114, change DO600to OD600

Response to point 9. Sorry for the mistake. Thank you.

  1. Line 186 – Why is the non-PAH control not shown and only a citation provided? 

Response to point 10. In the first version of the article we had not considered it necessary, but it has now been included in the Supplementary figure 1.

  1. In Fig. 7(b), wild type D. hansenii and S. cerevisiae BY4742 has similar or even higher CYP activity compared with the ones with DhDIT2 gene expression. Explain why BaP degradation was higher by S. cerevisiae BY4742 and S. cerevisiae dit2Δ-pYES2-DhDIT2? Does that mean the CYP activity is not important in BaP degradation?

Response to point 11. CYP activity in the wild-type strains is higher at day 0 (basal), then at day 2 in glucose than in the strains expressing the DhDIT2 gene, most likely due to regulation by glucose, as previously reported (Wiseman, et al., 1978). Furthermore, the expression and regulation of the DhDIT2 gene is different in the plasmid than in the ScDIT2 and DhDIT2 gene present in their respective chromosomes. In BaP, the activity is shown to be improved, compared to that of the respective strain without the DhDIT2 gene.

We apologize that our explanation was perhaps not clear enough in the Discussion section on these issues, we tried to improve it now. In Table 3, the degradation of BaP by S. cerevisiae dit2Δ pYES2-DhDIT2 should be compared to that of S. cerevisiae dit2Δ, with a difference of 9.8% to 46%, supporting the importance of this gene.  

Wiseman, A., Lim, T. K., & Woods, L. F. (1978). Regulation of the biosynthesis of cytochrome P-450 in brewer's yeast. Role of cyclic AMP. Biochimica et biophysica acta544(3), 615–623. https://doi.org/10.1016/0304-4165(78)90335-5

  1. Explain why this work evaluated temperature effects on BaP degradation only on wild type D. hansenii? Based on this study, wild type D. hanseniiwill not be further used in mycoremediation. The temperature effects should be studied on S. cerevisiae BY4742-pYES2-DhDIT2 as well.

Response to point 12. Actually, our proposal is to use both wild-type D. hansenii and S. cerevisiae BY4742 pYES2-DhDIT2 strains for mycoremediation, this is suggested at the end of the Discussion section. D. hansenii has the advantage of being of marine origin and would be suitable for use in ocean contaminated sites; our Fig. 3 (former Fig. 1), demonstrates the ability of this yeast to grow at low temperatures. We now added a supplementary figure (Fig. S3) with the experiment at 15 °C using S. cerevisiae BY4742 pYES2-DhDIT2, since 28 °C is the temperature at which the experiment in Fig. 7 was performed. This new figure can be compared with Figs. 3 and 7. An explanation is now included in the respective sections. We thank your suggestion.

  1. Figure 7 is very compelling!

Response to point 13. Thank you very much for this nice comment.

  1. The conclusion section is missing some perspectives related to future research work.

Response to point 14. You are right, thank you for the observation. We are now including also some work that is now being in course at the lab.

We would like to thank the reviewer for taking the time to make such a detailed review of our work so that we can improve it.

Reviewer 3 Report

1.    Have you assessed any ligninolytic enzymes activity of your strains growing in the BaP presence? Do you have any data related to the presence of ligninolytic enzymes in the genome of your strains and suggestion about their contribution to BaP transformation/degradation?

2.    DH-ALK1 gene was found in D. hansenii genome. Do you have any suggestions about involvement of alkane monooxygenase enzymes in the BaP transformation/degradation?

3.    I would strongly recommend to add results of HPLC or GC analysis for confirmation of BaP degradation by D. hansenii.

4.    Fig.1 and Fig.6 would be better to show in CFU calculation graph picture or table.

5.    Fig.2. How authors could explain increasing of specific growth rate in the presence of 500 ppm of BaP?

Author Response

Reviewer 3:

Open Review

English language and style

( )Extensive editing of English language and style required
( ) Moderate English changes required
( ) English language and style are fine/minor spell check required
(x) I don't feel qualified to judge about the English language and style

Yes

Can be improved

Must be improved

Not applicable

Does the introduction provide sufficient background and include all relevant references?

( )

(x)

( )

( )

Are all the cited references relevant to the research?

(x)

( )

( )

( )

Is the research design appropriate?

(x)

( )

( )

( )

Are the methods adequately described?

(x)

( )

( )

( )

Are the results clearly presented?

( )

(x)

( )

( )

Are the conclusions supported by the results?

( )

(x)

( )

( )

Comments and Suggestions for Authors

  1. Have you assessed any ligninolytic enzymes activity of your strains growing in the BaP presence? Do you have any data related to the presence of ligninolytic enzymes in the genome of your strains and suggestion about their contribution to BaP transformation/degradation?

Response to point 1. In multiple reviews it has been previously reported that ligninolytic enzymes are secreted by basidiomycetous fungi under ligninolytic conditions (Aranda, 2016). Of the four genera of yeasts analyzed in this article, only Rhodotorula mucilaginosa belongs to the phylum Basidiomycota and the other three to the phylum Ascomycota. A study of relative gene expression in R. mucilaginosa revealed the existence of a gene coding for a laccase enzyme, but the authors did not detect anything when trying to estimate the enzymatic activity (Peidro-Guzmán, et al., 2021). With respect to the other three yeast species, mainly in Debaryomyces hansenii using MycoCosm, no sequences were identified in its genome that would indicate genes coding for possible ligninolytic enzymes, so we did not consider it necessary to try to estimate the enzymatic activity of ligninolytic enzymes, in addition to the fact that our culture medium does not possess the ligninolytic conditions to induce these enzymes, in case they were found in its genome. However, this valuable comment led us to include in the introduction a small mention of the role of ligninolytic enzymes in the initial oxidation of BaP and other PAHs in fungi that can secrete them.

Aranda E. (2016). Promising approaches towards biotransformation of polycyclic aromatic hydrocarbons with Ascomycota fungi. Current opinion in biotechnology, 38, 1–8. https://doi.org/10.1016/j.copbio.2015.12.002

Martínez-Ávila, L., Peidro-Guzmán, H., Pérez-Llano, Y., Moreno-Perlín, T., Sánchez-Reyes, A., Aranda, E., Ángeles de Paz, G., Fernández-Silva, A., Folch-Mallol, J. L., Cabana, H., Gunde-Cimerman, N., & Batista-García, R. A. (2021). Tracking gene expression, metabolic profiles, and biochemical analysis in the halotolerant basidiomycetous yeast Rhodotorula mucilaginosa EXF-1630 during benzo[a]pyrene and phenanthrene biodegradation under hypersaline conditions. Environmental pollution (Barking, Essex: 1987)271, 116358. https://doi.org/10.1016/j.envpol.2020.116358

  1. DH-ALK1 gene was found in D. hansenii genome. Do you have any suggestions about involvement of alkane monooxygenase enzymes in the BaP transformation/degradation?

Response to point 2. In the new version of the manuscript, information related to DH-ALK1 has been added in the Discussion section. An exhaustive search of the NCBI database and others, allowed us to determine that the protein encoded by the DH-ALK1 gene is different from the sequence of the DhDIT2 gene.

  1. I would strongly recommend to add results of HPLC or GC analysis for confirmation of BaP degradation by D. hansenii.

Response to point 3. It was a little bit difficult to find where to take our samples being analyzed by GC-MS, and in the middle of our University vacations period, but finally, could manage a collaboration that was acknowledged at the appropriate section of our manuscript. This was the main reason of the delay in sending the final review but we are grateful for your recommendation; it was very valuable to improve this research. A GC-MS analysis was performed to confirm the degradation of BaP by D. hansenii, which allows us to corroborate the results obtained by fluorescence. Comments related to the results obtained by this technique are added in the results, and Supplementary Figure 2 includes some representative chromatograms.

  1. Fig.1 and Fig.6 would be better to show in CFU calculation graph picture or table.

Response to point 4. According to the literature (Kwolek-Mirek & Zadrag-Tecza, 2014), determining CFU or performing a spotting test provides the same information, i.e., analyzing cell viability. The objective of those experiments was to show a qualitative result; the quantitative ones are shown in Fig. 4 with the kinetics parameters calculated from the growth curves, and Fig. 5, where we measured the viability with alamarBlue. Therefore, we apologize for not carrying out the CFU determination, as the information obtained will be similar to that obtained with the methodology used.

Kwolek-Mirek, M., &Zadrag-Tecza, R. (2014). Comparison of methods used for assessing the viability and vitality of yeast cells. FEMS yeast research, 14(7), 1068–1079. https://doi.org/10.1111/1567-1364.12202

  1. Fig.2. How authors could explain increasing of specific growth rate in the presence of 500 ppm of BaP?

Response to point 5. This information has already been included in the article new version, thank you for the comment, as it enriched the results section.

We would like to thank the reviewer for taking the time to make such a detailed review of our work so that we can improve it.

Round 2

Reviewer 1 Report

I appreciate very much the efforts that the authors have devoted to improving their manuscrip, it may be accepted after the minor changes.

Line36,  a space at the beginning of the sentence is needed.

Fig. 4a,b, the initial OD600 of medium in the part of was 0.03, so why the log values of OD600 at 0 h were higher than 0.15? The explanation at lines 352- 354 was not clear.

Line 597, the degradation function of DhDIT2 gene was verified by transforming the gene DhDIT2 into S.cerevisiae BY4742, does the strain S.cerevisiae BY4742 itself have the ability to degrade BaP or not?

Author Response

Reviewer 1, Round 2.

I appreciate very much the efforts that the authors have devoted to improving their manuscript, it may be accepted after the minor changes.

Thank you very much for your suggestions, they were really helpful to improve our work.

Line36, a space at the beginning of the sentence is needed.

Thank you for noticing it. We have added the space at the beginning of the paragraph

Fig. 4a,b, the initial OD600 of medium in the part of was 0.03, so why the log values of OD600 at 0 h were higher than 0.15? The explanation at lines 352- 354 was not clear.

In lines 352-354, and in section 2.2 of Materials and methods, it is explained that the initial suspension was adjusted to an OD600=0.03 in our lab spectrophotometer (Beckman DU500), and when this is set into the multi-well plate and inserted in the Bioscreen C automatic plate reader, when the OD600 is selected to be read, the initial reading is around 0.2 for all the samples except the ones from 500 ppm BaP (this is also explained in the same paragraph). The obtained values in the plate reader were then graphed in excel and the y-axis scale was selected to be displayed as a log scale. We added a reference to section 2.2 in lines 355-356 and in section 2.2, also added a few words trying to make it well-defined.

Line 597, the degradation function of DhDIT2 gene was verified by transforming the gene DhDIT2 into S. cerevisiae BY4742, does the strain S. cerevisiae BY4742 itself has the ability to degrade BaP or not?

We tried to make clearer this paragraph, adding extra information. The wild-type BY4742 has the ability to degrade BaP (Tables 2 and 3), however, it was of our interest to see if regardless the presence of ScDIT2 gene, if transforming it with an extra copy, now of DhDIT2, there was an improvement in the BaP degradation capacity, and this finally could be demonstrated (Table 3).

We want to express our gratitude to all the reviewers for their time in making better our manuscript.

Reviewer 2 Report

All my previous comments have been adequately addressed by the authors.  So recommend acceptance of this manuscript.  Thank you. 

Author Response

We appreciate the time and excellent suggestions made by reviewer 2. 

Our best regards,

The authors.